# A machine learning-based chemoproteomic approach to identify drug targets and binding sites in complex proteomes

Ilaria Piazza [1,2,3,6], Nigel Beaton[2,6], Roland Bruderer[2], Thomas Knobloch[4], Crystel Barbisan[4], Lucie Chandat[4], Alexander Sudau[4], Isabella Siepe [5], Oliver Rinner[2], Natalie de Souza[1], Paola Picotti[1,7 ✉] & Lukas Reiter [2,7 ✉]

Chemoproteomics is a key technology to characterize the mode of action of drugs, as it directly identifies the protein targets of bioactive compounds and aids in the development of optimized small-molecule compounds. Current approaches cannot identify the protein targets of a compound and also detect the interaction surfaces between ligands and protein targets without prior labeling or modification. To address this limitation, we here develop LiP-Quant, a drug target deconvolution pipeline based on limited proteolysis coupled with mass spectrometry that works across species, including in human cells. We use machine learning to discern features indicative of drug binding and integrate them into a single score to identify protein targets of small molecules and approximate their binding sites. We demonstrate drug target identification across compound classes, including drugs targeting kinases, phosphatases and membrane proteins. LiP-Quant estimates the half maximal effective concentration of compound binding sites in whole cell lysates, correctly discriminating drug binding to homologous proteins and identifying the so far unknown targets of a fungicide research compound.

---

[1] ETH Zürich, Institute of Molecular Systems Biology, Department of Biology, Otto-Stern-Weg 3, 8093 Zürich, Switzerland. [2] Biognosys AG, Wagistrasse 21, 8952 Schlieren, Switzerland. [3] Max Delbrück Center for Molecular Medicine, Robert-Rössle Straße 10, 13125 Berlin, Germany. [4] Bayer SAS, Crop Science Division, Lyon, France. [5] BASF SE, Ludwigshafen, Germany. [6] These authors contributed equally: Ilaria Piazza, Nigel Beaton. [7] These authors jointly supervised this work: Paola Picotti, Lukas Reiter. ✉email: picotti@imsb.biol.ethz.ch; lukas.reiter@biognosys.com

Unraveling the mechanism of action and molecular target of small molecules remains a major challenge in drug development[1]. Knowledge of the direct target of a drug is essential for devising strategies to modulate the compound's effects. Recently, new proteomics approaches have enabled the investigation of protein–drug interactions in native environments[2–4]. One such strategy relies on the use of chemical probes, which allow the enrichment and the identification of protein targets and may include the interaction sites as well[5]. However, these target capturing probes may perturb molecular interactions and biological functions. Complementary strategies such as thermal proteome profiling (TPP)[6–8], stability of proteins from rates of oxidation (SPROX)[9] and drug affinity responsive target stability (DARTS)[10–12] map drug interactions by assessing variations of thermal stability, susceptibility to oxidation or to proteolytic degradation induced by ligand binding. These methods bypass the need for drug or target modification that may create artifacts but might not detect the most low-abundant protein targets, due to the lack of an enrichment step. Thus, drug target characterization remains a challenge. Especially lacking are approaches that map interactions on a proteome-wide scale without requiring drug labeling, and that identify the drug binding site with peptide level resolution.

We recently used limited proteolysis (LiP) to map metabolite binding proteins directly in whole cell lysates of microbial organisms (LiP-SMap). This is achieved through the global detection with liquid chromatography-coupled tandem mass spectrometry (LC-MS) of differential proteolytic patterns produced upon ligand binding[13]. However, this approach was only used and validated for the study of microbial organisms with proteomes of limited complexity.

Here we adapt this LiP-based approach to enable the systematic investigation of protein–small-molecule interactions in complex eukaryotic proteomes (e.g. human). To do so, we employ a machine learning-based framework (LiP-Quant) that makes use of drug dose titrations, identifies small-molecule compound targets and can provide additional information including binding site prediction and target affinity. To evaluate its performance, we focus on the identification of drug targets, an application of particular breadth and interest. We benchmark LiP-Quant by investigating binding of compounds to known human protein targets, addressing the possibility of detecting kinase or phosphatase inhibitors and membrane protein drugs. We also identify an unknown target of a research fungicide currently under study.

## Results

**Identification of human drug targets by limited proteolysis.** First, we tested the applicability of the LiP workflow in native human cellular environments with the drug rapamycin. Rapamycin is a well-characterized drug known to interact with only one protein, FKBP1A (FK506-binding protein 1, also known as FKBP12)[14,15]. Two approaches were taken to test target identification, treatment of HeLa cell lysates as well as direct treatment of live HeLa cells prior to lysis, with 2 μM rapamycin or vehicle in two independent experiments. Proteome extracts were then subjected to limited proteolysis (LiP) under controlled conditions[13] (Supplementary Fig. 1A). Differentially abundant proteolytic peptide fragments between rapamycin and vehicle-treated samples were identified on a proteome-wide scale using a label-free Data Independent Acquisition Mass Spectrometry (DIA-MS) approach[16]. We quantified 5265 protein from HeLa cell lysates and 6068 from HeLa cells. After applying filtration criteria based upon relative peptide abundance changes and fixed statistical significance thresholds, we identified 52 putative drug targets with at least one differentially abundant peptide in lysates and 37

candidate proteins from the treatment of live cells, including the known target FKBP1A in both cases (Fig. 1a) (Supplementary Data 1, 2). Since rapamycin is known to interact with only one protein, FKBP1A, but we identified targets in addition to FKBP1A in both experiments, we realized that this workflow with one drug dose may not discriminate unequivocally true protein interactors and false positive identifications. This is likely due to the high complexity of the human proteome and we therefore sought an expanded approach that emphasizes target prioritization amongst potential targets identified.

**LiP-Quant identifies drug targets via machine learning.** To address this challenge, we devised the LiP-Quant workflow for the deconvolution of direct drug targets in the human proteome, a method that has been optimized to reduce the noise observed in single drug dose LiP assays. In this pipeline, protein lysates are exposed to a compound dosage dilution series followed by limited protease cleavage with proteinase K. Upon compound binding proteolytic patterns of drug targets should be altered, with true target peptides showing a change in abundance proportional to drug concentration (Fig. 1b). In establishing LiP-Quant we focused on drug-treated HeLa cell lysates, since direct physical interactions between a target protein and a ligand are more effectively found in cell extracts rather than living cells[6,7,17–19]. We reasoned that the knowledge of primary protein targets might be more beneficial than elucidating downstream pathways engaged by the drug (which could be found with assays in vivo) at an early stage of the drug-discovery pipeline.

We used machine learning to derive peptide attributes that prioritize true drug target identification and built a composite score (LiP-Quant score) by which these peptides can be ranked. Four features were identified that contribute to a target peptide's LiP-Quant score with the dominant component being correlation ($R^2$) to a sigmoidal trend of the drug dose–response profile (69% of the LiP-Quant score weight, LiP-Quant score component I) (Supplementary Fig. 1B). The frequency of a protein's identification in experiments where it is not a confirmed target (protein frequency library, PFL) was also considered. Within the PFL score, proteins received more weighting the less frequently they were known to contaminate other experiments (LiP-Quant score component II). In addition, multiple peptides with high dose–response correlations from the same protein (LiP-Quant score component III), and calculated significance (q-value) of differential peptides between the vehicle and three concentrations above the compound's known EC50 (LiP-Quant score component IV) were found to contribute to target peptide identification at approximately 10% of the LiP-Quant score each (Supplementary Fig. 1B).

In total, six ground truth compounds, i.e. the drug target(s) is known, were used to ascertain the properties of peptides from true targets, including how much each attribute contributes to correct target identification (Supplementary Figs. 1B, 2A). Each of these four LiP-Quant score components contributes to the predictive power of the approach, outperforming either experiments with only one drug dose (LiP-SMap) or dose–response correlation alone (LiP-Quant score component I) (Supplementary Fig. 2B). We observed that LiP-Quant scores show a bimodal distribution across the positive control datasets with target protein peptide scores clearly enriched in the high-scoring peak of the distribution. Therefore, we defined putative target peptides as those with a LiP-Quant score above 1.5, which is the median score of non-target peptides plus three standard deviations (Fig. 1b, Supplementary Fig. 3, Methods section: guidelines for interpreting LiP-Quant results). This threshold equates to an average positive predictive value (PPV) of 30% across positive

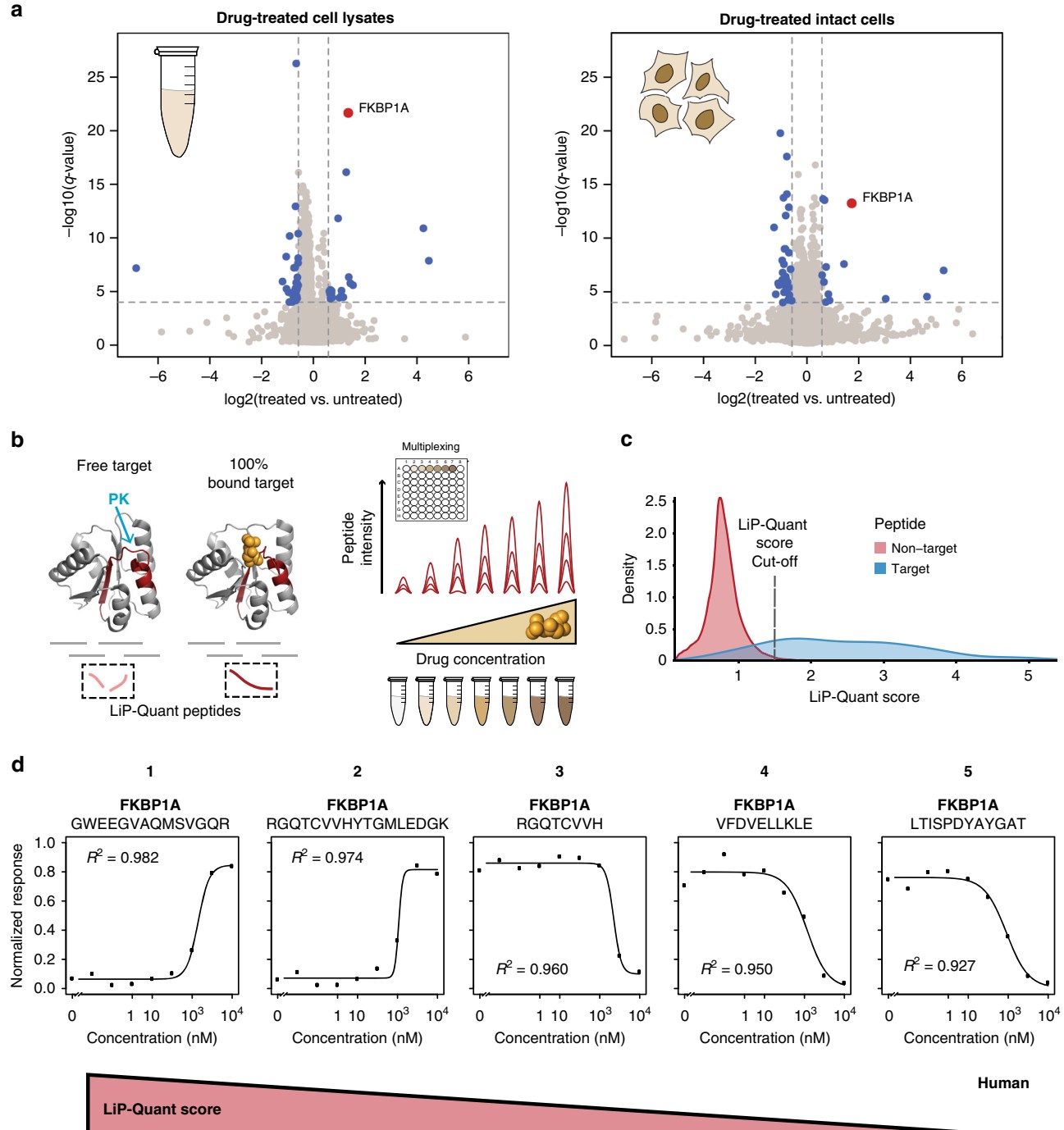

**Fig. 1 LiP-Quant, a platform for drug target identification. a** Volcano plots of LiP samples from HeLa lysates (left panel, $n = 4$ of independent lysate replicates) and live HeLa cells (right panel, $n = 3$ of biologically independent cells) treated with rapamycin. Peptide mixes produced in the presence or absence of 2 μM rapamycin are compared. Fold changes (FC) in peptide abundance for treated and untreated samples are shown as a function of statistical significance. Significance cutoffs were $q$-value = 0.0001 ($t$-test, one sample, two-tailed) and FC = 1.5. Each protein is represented with a single data point, corresponding to the peptide with the lowest $q$-value. The known interactor of rapamycin (FKBP1A) is highlighted in red. Proteins passing both cutoffs are in blue. **b** Principle and experimental design of LiP-Quant. Sample preparation for MS analysis follows a multiplexed workflow that is suitable for the processing of drug libraries. **c** Compiled LiP-Quant score distributions (Gaussian smoothed kernel density) for known target and non-target proteins from all HeLa experiments with the exception of the promiscuous binder staurosporine. **d** Dose–response curves showing relative intensities of LiP-Quant peptides after partial proteolytic digestion of aliquots of HeLa lysates over a rapamycin concentration range. Curves of the top 5 LiP-Quant peptides ranked by LiP-Quant score, all of which are from the expected direct target FKBP1A, are shown. The numbers on top of the graph show the LiP-Quant score position of the relative peptides. Source data are provided as a Source Data file.

control experiments, although higher PPVs can be achieved by applying more stringent thresholds (e.g. the top 10 peptides yields an average PPV of 70%) (Supplementary Fig. 2C).

After validation, we tested the LiP-Quant pipeline with rapamycin in HeLa lysates using a project specific spectral library of 8213 proteins. Of the 110,668 peptides mapping to 5295 proteins quantified by DIA-MS, multiple LiP-Quant peptides, including the 5 top-scoring peptides, mapped to FKBP1A (Fig. 1d) (Supplementary Data 3). This made FKBP1A the highest-ranking candidate protein target and showed the ability of LiP-Quant to identify correct drug targets. To further test the LiP-Quant pipeline we selected a different compound, FK506, which is also known to target FKBP1A. Here, we obtained very consistent results to those of rapamycin as again the top 5 LiP-Quant peptides mapped to FKBP1A (Supplementary Fig. 4A). Interestingly, target peptide identification across the experiments was very robust with 7 of the same LiP-Quant peptides identified in both experiments (Supplementary Data 3). In each case the peptide showed the same directionality of regulation (i.e. up or down regulation upon compound binding) and had similar LiP-Quant scores, demonstrating the consistency of the approach (Supplementary Data 4).

From this, we concluded that the LiP-Quant approach is suitable for drug-target deconvolution experiments in complex human proteomes and that ranking peptides by LiP-Quant score enables prioritization of genuine targets from random hits and reduces false positive identifications.

**Applicability of LiP-Quant to other eukaryotic species**. Since rapamycin's target is conserved across species, we exploited this to test LiP-Quant more generally in species other than *H. sapiens*. In a rapamycin LiP-Quant experiment with *S.cerevisiae* lysates the 5 top-scoring peptides (of a total of 30,209 peptides and 2553 protein groups identified), mapped to FRP1, the known target in *S.cerevisiae* (Supplementary Data 5) (Supplementary Fig. 4B, C), showing the equivalence of the approach in yeast and humans.

Beyond the known target of rapamycin (FRP1) in yeast, we identified the previously uncharacterized potential targets, ARI1 and SYEC, with high LiP-Quant scores (>2.5) (Supplementary Fig. 4D). These proteins could represent alternative binders (off-target effects) or undergo secondary structural effects upon TOR1 pathway activation in cell lysate. To discriminate between these two cases, the same LiP-Quant experiment was performed with lysates where TOR signaling was impaired via a mutation in the *tor1* gene and a deletion of the *fpr1* gene *(tor1-1 Δfpr1)*[20]. Thus, in this system we should not detect structural changes downstream of TOR activation. In the *tor1-1 Δfpr1* proteome, only peptide GDLVITEESWNK of ARI1 was detected as a hit. From this, we conclude that ARI1 is likely a secondary target of rapamycin (Supplementary Fig. 4E, F) (Supplementary Data 5) and that LiP-Quant preferentially detects direct drug binding proteins.

**Benchmarking LiP-Quant for target identification**. To further validate LiP-Quant we tested the method with staurosporine, a universal kinase inhibitor (KI)[21,22]. This compound has been previously assayed with TPP[6] and bead-immobilized kinase inhibitor (kinobeads) pulldowns[23], thus it enables comparative analysis among these three chemoproteomic methods.

We ranked the protein candidate targets using LiP-Quant scores and the corresponding scoring variables of TPP and kinobeads (see Methods section) considering the 512 annotated human protein kinases in KinHub (http://www.kinhub.org) as true positives (Supplementary Data 6). LiP-Quant discriminates true positive targets amongst the top-ranking candidate protein targets of staurosporine (Fig. 2a). Overall, the number of true positive targets found by the LiP-Quant and TPP methods are comparable, although TPP is more sensitive than LiP-Quant as it identifies more kinase targets in total (21 and 49 found kinase targets respectively, Fig. 2b). Notably, the kinobeads method captures significantly more kinases than TPP and LiP-Quant (190 kinases), being an enrichment-based method (Fig. 2b, see Methods section). Only a limited number of kinases (6) were detected by all three LiP-Quant, TPP and kinobeads approaches (Fig. 2b) (Supplementary Data 7), showing that the approaches are complementary. This limited overlap is likely due to probing of rather distinct sub-populations of the kinome, as the three methods assess drug binding measuring different protein properties.

Within the LiP-Quant staurosporine dataset a higher median protein sequence coverage was observed among successfully detected kinase targets (Fig. 2c) when compared to the entire proteome but in particular relative to undetected kinases. This suggests that high sequence coverage is crucial for the identification of drug targets by LiP-Quant. From this we hypothesized that peptide under sampling (i.e. lower protein coverage) could lead to reduced kinase target identification, and thus attempted to increase proteome sequence coverage via longer LC-MS gradients (Deep LiP-Quant) on the same samples. Here we found that the performance of LiP-Quant for target identification improved, as the Area Under the Curve (AUC) of receiver operating characteristic curves increased from 0.76 to 0.81 (Fig. 2d), identifying 42% more kinases than standard LiP-Quant with an increase from 21 to 36 found kinase targets (Supplementary Data 7 and 8). The sensitivity of TPP could also be improved at the cost of increased MS instrument time by measuring variations of thermal stability over a drug dose and temperature gradients[24]. As recently reported[25], analyzing TPP data with a non-parametric approach increases specificity and sensitivity of TPP assays. We tested TPP in combination with non-parametric statistical analysis and observed that this has the highest predictive power for staurosporine targets (Fig. 2d). Overall, these results demonstrate that LiP-Quant is a comparable and complementary approach to TPP for the detection of drug binding proteins without requiring prior modifications of the drug. While LiP-Quant might be limited by protein sequence coverage, TPP is instead insensitive to targets that do not change their melting behavior upon drug binding.

LiP-Quant may be biased against membrane proteins, since proteomes are extracted and the insoluble proteins removed by centrifugation prior to LiP (Supplementary Fig. 1A) (see Methods section). Despite this, approximately 200 of the quantified proteins (typically 4% of the total) are annotated as cell surface proteins that reside in the plasma membrane[26]. When we applied LiP to crude lysates without centrifugation, as used for the protocol in drug-treated live cells (Supplementary Fig. 1A), membrane associated protein identification increases to more than 300 proteins. Using a plasma membrane enrichment approach, we were able to identify approximately 400 membrane associated proteins, a 100% enrichment from standard LiP-Quant (Supplementary Fig. 5A) (see Methods section). Using this plasma membrane proteome extraction protocol, we successfully identified the alpha subunit of the sodium/potassium-transporting plasma membrane pump (ATP1A1) as the known target of the drug proscillaridin A (Supplementary Fig. 5B, C) (Supplementary Data 9) by improving the coverage of membrane proteins. A single concentration rapamycin control experiment confirmed that the addition of mild detergent during cell lysis does not disturb compound target binding (Supplementary Data 10).

These results suggest that the sensitivity and coverage of LiP-Quant will improve by reducing proteome under sampling, which

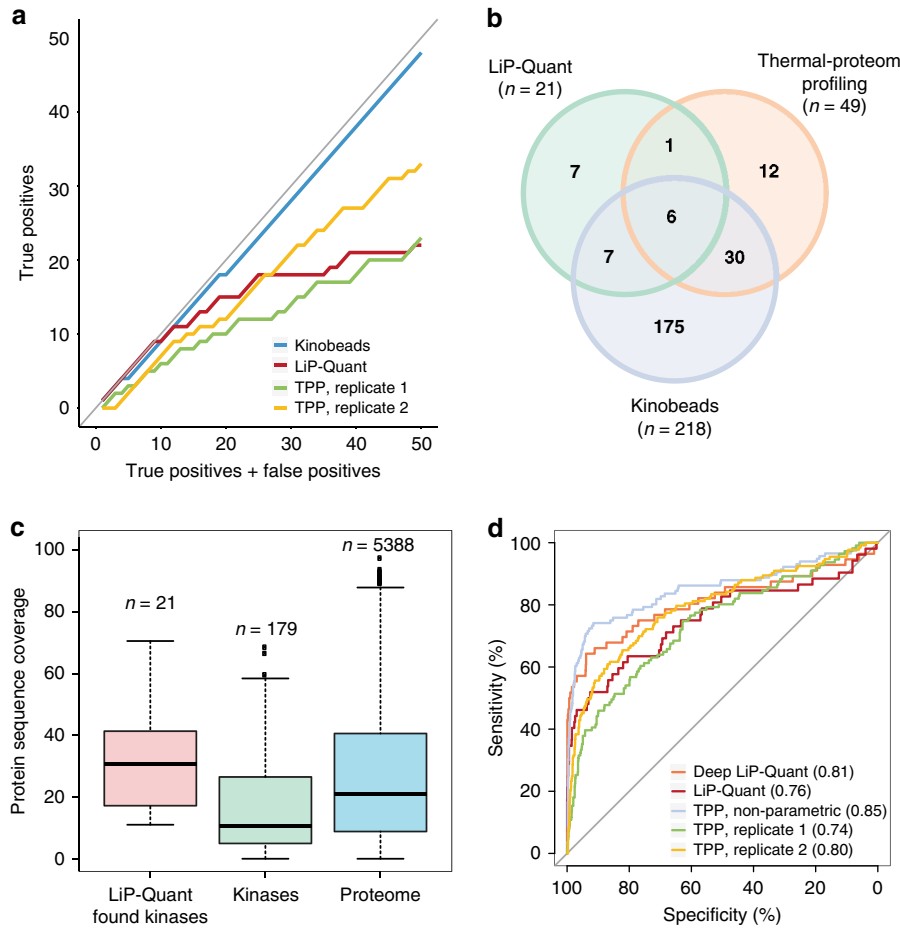

**Fig. 2 Benchmarking the LiP-Quant approach. a** True positive rate evaluation for selected assays (LiP-Quant, thermal proteome profiling (TPP) and kinobeads) focused on kinase target identification for staurosporine[6,23]. True positive hits in the top 50 candidates (see Methods section) are shown as a function of the number of true and false positives in the candidate list for the respective assays. Since staurosporine is a promiscuous binder of protein kinases, we considered the entire protein kinase space as the reference for true positive identifications. The gray line indicates a perfect candidate list containing only true positives (slope = 1). **b** Total number of kinase targets, as well as common kinase targets of staurosporine found in the LiP-Quant, TPP and kinobeads experiments. **c** Distribution of protein sequence coverage for kinases identified by LiP-Quant (pink, $n = 21$), all quantified protein kinases by MS in HeLa cells (green, $n = 179$) and the entire measured proteome (blue, $n = 5388$). In the box plots the central line defines the median, the bounds of box the first and third quartile, the whiskers are the minimal and maximum values. **d** Receiver operating characteristic (ROC) curves of staurosporine protein interactions measured by three different chemoproteomics methods. The ROC curve of LiP-Quant (red) is compared with that of two replicates of TPP experiments based on melting point fittings (green and yellow), a non-parametric analysis of whole TPP dose–response curves (cyan) and LiP-Quant with a LC-MS gradient of 4 h (Deep LiP-Quant). Bracketed numbers represent areas under the curves (AUC). The gray line represents a random classifier. The ground truth is represented by the 512 protein kinases present in the human genome. Source data are provided as a Source Data file.

is typically achieved by deeper MS analyses or fractionation techniques.

**Quantitative parameters of kinase and phosphatase inhibitors.** Next, we focused on kinases and phosphatases as drug targets of particular pharmacological interest, given their frequent dysregulation in disease, particularly in cancer. One common issue with these classes of targets is their variable selectivity, as some of these drugs have more than one protein target in the cell, and frequently bind them with different affinities[27], thus it is important to characterize these subtle differences to determine the specific modes of actions. We reasoned that in LiP-Quant experiments we could determine drug binding affinity directly from cell extracts by utilizing the concentration of drug at which we observe a variation of 50% of the maximum LiP-Quant signal (EC50).

First, we studied the kinase inhibitor selumetinib. We identified 6 peptides that map to MAPK2K1 and MAP2K2, the known

targets of the drug. We also identified NQO2, a confirmed off-target of other kinase inhibitors[21,28,29] (Supplementary Data 3), as a potential target of selumetinib. This protein was also identified as a candidate target protein for staurosporine, further supporting its off-target binding ability with respect to kinase inhibitors. Unlike staurosporine (Fig. 2), which LiP-Quant characterizes as a broad kinase inhibitor, selumetinib was correctly profiled as a specific MAPK inhibitor, indicating that LiP-Quant can recapitulate differential selectivity profiles (Supplementary Fig. 6A, B). We further investigated the ability of LiP-Quant to parse binding events for highly homologous protein targets by focusing on compounds that inhibit specific members of the serine/threonine phosphatase (PP) family (Supplementary Fig. 6C). As expected, LiP-Quant peptides for calyculin A all map to either PP1 or PP2A/B, while the fostriecin assay only identified peptides from PP2A/B, PP4 and PP6 as targets (Supplementary Data 3, 11).

The EC50 values we extracted for selumetinib and MAPK2K1 vary between 48.5 and 101 nM, which are slightly above the EC50

of 41 nM measured with alternative methods (Supplementary Fig. 7A)[30]. Further, we identified 12 LiP-Quant peptides from PP2A/B and 16 from PP1 in the calyculin A experiment. Their EC50s across peptides for a given target are largely comparable, with median EC50 of 18 nM and 63 nM, respectively (Supplementary Data 3). These values are approximately 10-fold higher than those measured in vitro (Supplementary Fig. 7B)[31]. However, the ratio between the calyculin A EC50 inferred by LiP-Quant for PP2A/B and PP1 closely reflects the previously reported 3.5-fold EC50 ratio difference between PP2A and PP1[30,32,33]. This ability of LiP-Quant to approximate absolute EC50 values and to effectively discriminate relative affinities between drug targets should help determine preferential target proteins of compounds (Methods section: Guidelines for interpreting LiP-Quant results). These experiments demonstrate that LiP-Quant can be used to successfully discriminate the binding promiscuity of compounds that target highly homologous proteins, even when there are very subtle differences in protein structure and compound specificity.

**LiP-Quant peptides as proxy for drug binding site positions**. LiP-Quant uniquely identifies interactions at peptide level resolution. As previously observed for metabolite binding[13], the positions of LiP-Quant peptides are typically in very close proximity to the known drug binding sites (Supplementary Fig. 8A). For instance, calyculin A LiP-Quant peptides from PP2A/B overlap with the conserved calyculin A binding cleft (Supplementary Fig. 8B, C). Based on this observation we used the peptide level information intrinsic to LiP-Quant to provide a proxy for the position of drug binding sites. We considered the three highest-scoring LiP-Quant peptides of known drug targets measured with rapamycin, FK506, selumetinib, staurosporine, fostriecin and calyculin A in HeLa proteome extracts (see Methods section). We then compiled those protein–drug complexes where a protein structure was available, which totaled seven proteins from all experiments. For each we calculated the center of mass of the atoms of these three LiP-Quant peptides to use as a reference for the approximate position of compound binding sites. The center of mass was represented as a point with geometric coordinates in the protein structure of these protein–drug binary complexes (Fig. 3). Among all known cases analyzed, with the exception of calyculin A and fostriecin, the distance between the amino acids surrounding this reference point and the drug ligand were within the Van der Waals distance of 4 Å (Fig. 3), unlike when three random peptides were sampled (Supplementary Data 12). Based upon these observations we concluded that we could use our center of mass triangulation approach to estimate drug binding sites.

**Characterization of a research fungicide using LiP-Quant**. Given its abilities to identify and characterize drug targets and their binding sites, we applied LiP-Quant to a research fungicide compound with an unknown target (BAYE-004, Fig. 4a). This lead compound was previously found to inhibit cell growth of *Botrytis cinerea*, a mold crop parasite (Fig. 4b). Using LiP-Quant, we identified two kinases as potential targets of BAYE-004, including Bcin06g02870 (Fig. 4c), predicted to be the *B. cinerea* homolog of casein kinase I. Two LiP-Quant peptides from Bcin06g02870 have a low nanomolar EC50 of 6 and 5 nM (LiP-Quant rank #6 and #7), approximately 1000-fold lower than the EC50 of the rest of the LiP-Quant peptides above the 1.5 threshold (Supplementary Data 13). We therefore tested further whether Bcin06g02870 was the primary direct target of the compound, using a *B. cinerea* cell line expressing a His-tagged version of Bcin06g02870. Cellular thermal shift assays (CETSA)[34]

demonstrated that this protein is thermally stabilized in vivo upon treatment with BAYE-004, confirming compound binding to Bcin06g02870 (Fig. 4d).

We calculated the center of mass of the #6 and #7 ranking LiP-Quant peptides to estimate the position of the BAYE-004 binding site within Bcin06g02870. Here we considered two LiP-Quant peptides for the center of mass calculation, as there were only two high scoring LiP-Quant peptides that could be used. The putative binding site of the drug maps in the ATP-binding site of Bcin06g02870, which is also a common binding site for kinase inhibitors (Fig. 4e, Supplementary Data 3). Finally, we performed a LANCE Ultra kinase assay (Perkin Elmer) to measure the phosphorylation activity on a peptide substrate of a purified MBP-tagged version of Bcin06g02870. BAYE-004 was found to inhibit Bcin06g02870 activity with a low nanomolar IC50 of 12.5 nM, confirming the relevance of the target identified (Fig. 4f).

The second putative target found, Bcin16g04330 is a predicted kinase homolog of glycogen synthase kinase-3β (LiP-Quant rank #1, #3 and #4) (Supplementary Data 13) (Supplementary Fig. 9A). This kinase is likely not the primary target of BAYE-004 as the extrapolated EC50 is several orders of magnitude higher than that of Bcin06g02870 (Fig. 4c, f). Interestingly, the center of mass of the LiP-Quant peptides ranking #1, #3, and #4 is situated in a domain containing an allosteric site in homolog kinases of Bcin16g04330[35,36]. This suggests that the measured structural effect results from binding to a secondary site or from compound binding-induced conformational changes of allosteric nature (Supplementary Fig. 9B, Supplementary Data 13).

Taken together, we have used LiP-Quant to identify direct binding targets of a previously uncharacterized research fungicide. Our data are consistent with BAYE-004 inhibition of fungal cell growth (Fig. 4a) via inhibition of the *B. cinerea* homolog of casein kinase I, providing a potential explanation for the previously unknown mode of action of the drug.

## Discussion

Devising modification-free chemoproteomic strategies that simultaneously probe whole-proteomes will increase efficiency in the drug development pipeline. Here, we presented LiP-Quant, a machine learning-based method that deconvolutes altered proteolytic patterns upon drug binding in complex proteomes. LiP-Quant substantially enriches for direct drug targets in both human cell lines and yeast with similar efficiency as other proteome-wide-scale techniques[6,21,37,38]. Although overall the TPP-based approaches examined here are more sensitive, it should be noted that true positive identification rates for LiP-Quant and TPP are virtually identical for the first 25 targets identified. Further, LiP-Quant differs from TPP-based approaches as it has peptide level resolution without requiring chemical modifications of drug compounds, which allow the identification of drug binding sites without a bias towards specific reaction chemistries or residue conservation.

LiP-Quant identifies peptides that undergo structural changes upon compound binding, providing a proxy for drug binding sites and enabling a quantitative assessment of their binding specificity and selectivity through the estimation of EC50s. This ability to accurately identify relative binding affinities for proteins, in particular among closely related protein families, is a highly beneficial feature when characterizing and refining drug leads. For instance, we used LiP-Quant with additional validation to discover the potential targets of a research fungicide. Interestingly, LiP-Quant suggested that the compound binds strongly (in the nM range) at the predicted ATP-binding pocket of a casein kinase-1 homolog, a mechanism consistent with many well characterized kinase inhibitors.

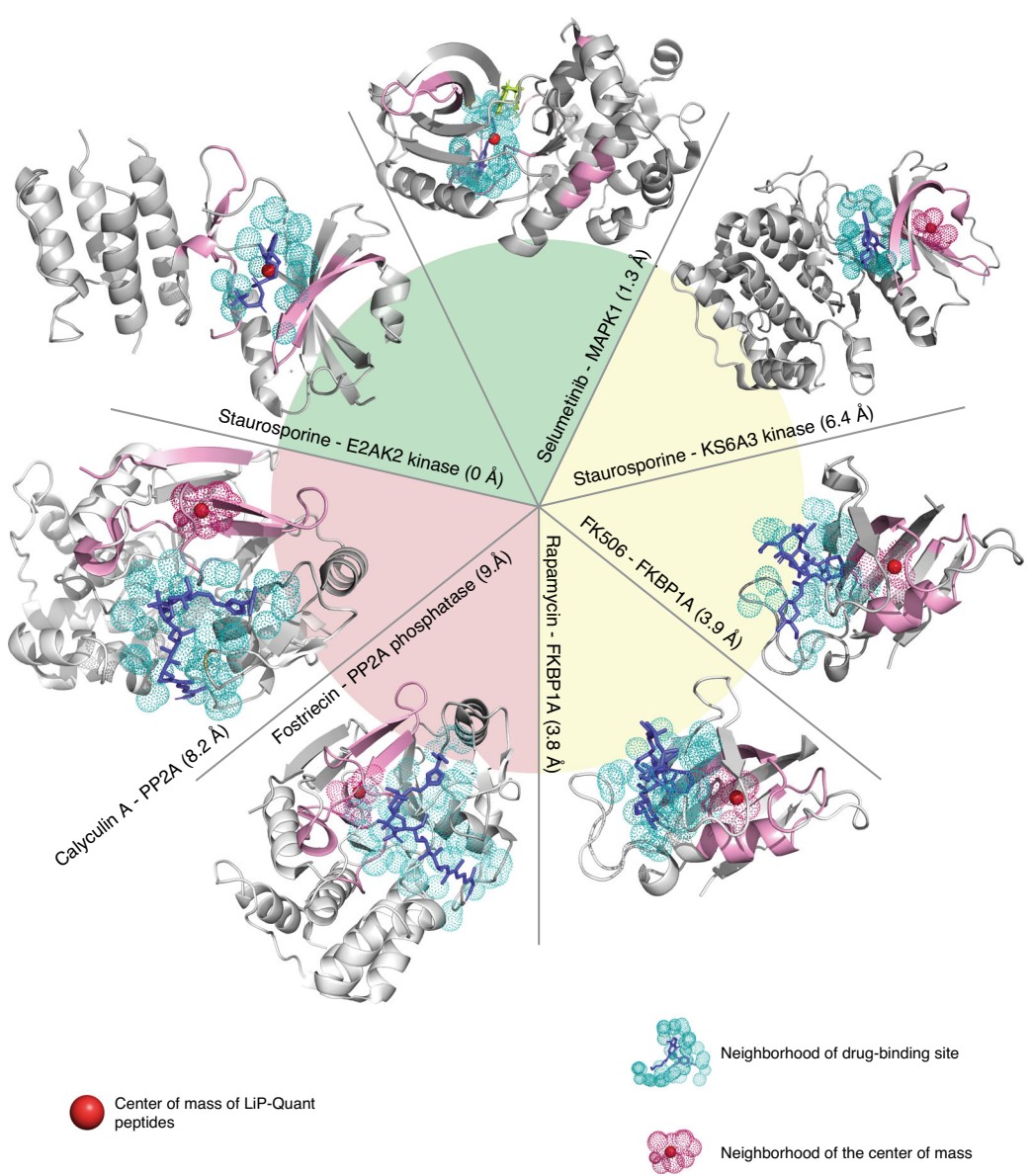

**Fig. 3 Approximating drug binding sites positions with LiP-Quant.** Structures of protein–drug complexes: (clockwise sorting) Staurosporine—E2AK2 kinase (PDBID: 2a19), Selumetinib— MAPK2K1 (PDBID: 4u7z), Staurosporine—K66A3 kinase (PDBID: 4nus), FK506–FKBP1A (PDBID: 1fkj), Rapamycin— FKBP1A (PDBID: 2dg3), Fostriecin—PP2A (PDBID: 1it6), Calyculin A—PP2A (PDBID: 1it6). Drug ligands are depicted in blue and LiP-Quant peptides in pink. The center of mass of the LiP-Quant peptides in each structure is shown with red spheres. Cyan shaded dots show a 4 Å radius (equivalent to van der Waals distance) from the drug ligand atoms and are used as a proxy for the residues of the ligand binding cleft. Magenta shaded dots show a 4 Å radius (equivalent to van der Waals distance) from the residues at the center of mass of the LiP-Quant peptides. Minimal distances between the LiP-Quant peptide center of mass and the drug binding cleft are reported. The green and yellow wedges indicate the drug protein pairs where the drug binding site and center of mass overlap, or show intersecting volumes, respectively. Calyculin A and Fostriecin (beige slice) are the only two cases where the ligand center of mass neighborhood and the drug binding site do not converge, although they are proximal.

We have found that LiP-Quant detects interactions with drugs over a broad range of affinities from nM to µM. EC50s estimated from lysate with LiP-Quant are often higher, by approximately an order of magnitude, than literature-reported values generally measured with recombinant proteins. The observed differences in EC50 values could thus simply be due to competition between different targets that occur in the lysate but not in vitro, or due to the effects of PTMs, protein—protein interactions, binding of other small molecules or the presence of membranes. The EC50 estimated with LiP-Quant may in fact be a more physiological indicator of drug–target binding affinity, since they are measured directly in complex biological mixtures.

Despite the absence of target enriching chemical probes, with LiP-Quant we detect approximately 70% of the proteins expressed in HeLa cells[39]. We can further increase proteome coverage with deeper MS analyses via longer LC-MS gradients and protein enrichment of whole cell lysates. In both cases an increase in protein coverage was observed within the compound protein target space (e.g. surface membrane proteins[26] for proscillaridin A and among kinases for staurosporine) and experimental results were concurrently improved.

We applied the LiP-Quant pipeline with cell extracts to prioritize the discovery of direct drug interactions. Further applying the LiP-Quant pipeline to cells that remain intact during

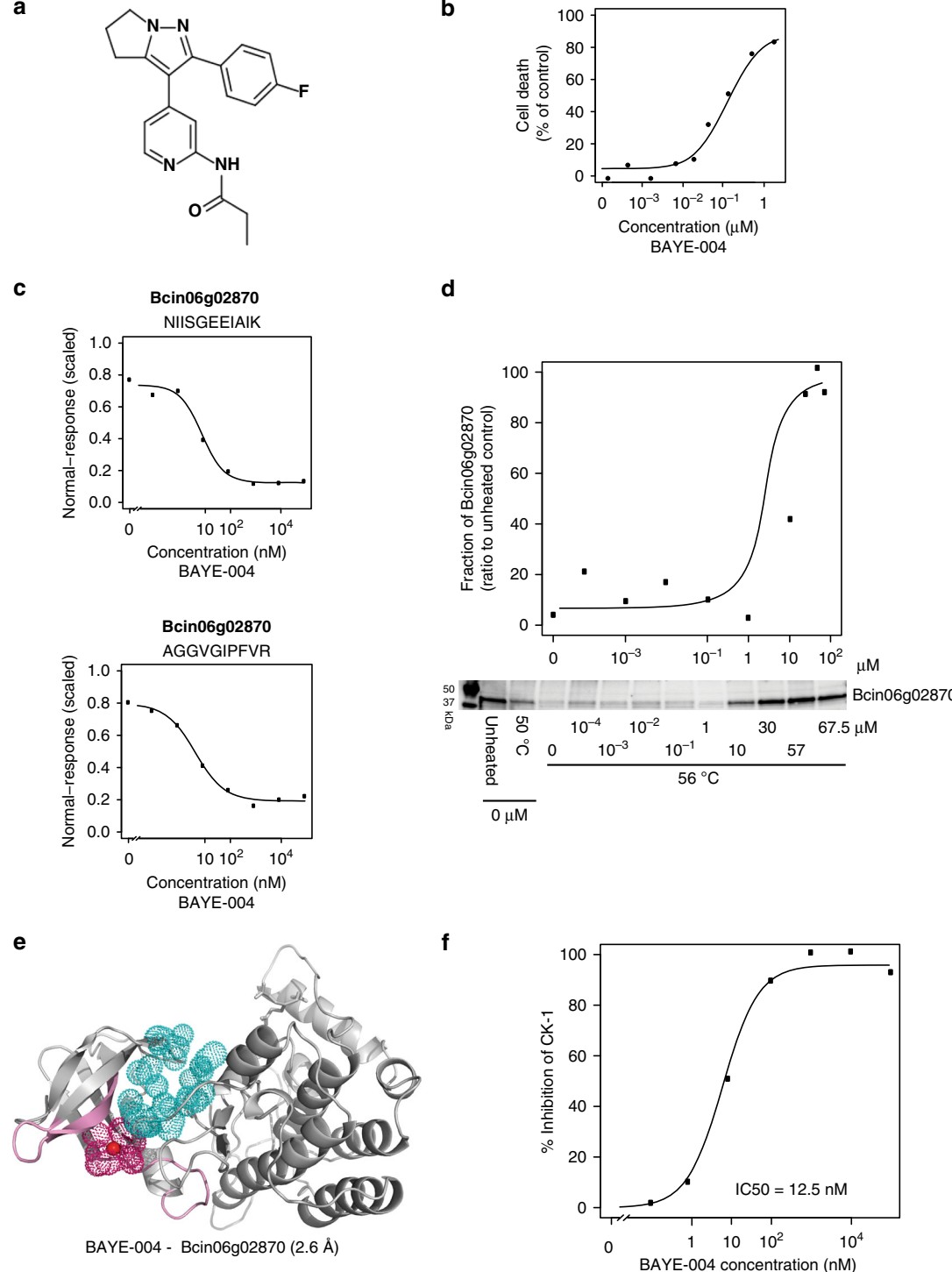

**Fig. 4 Identification of the kinase target and its binding site of a fungicide. a** Chemical structure of the novel compound BAYE-004. **b** Inhibition of growth in *Botrytis cinerea* cells upon treatment with increasing concentrations of BAYE-004. **c** Dose–response curves showing relative intensities of the top two LiP-Quant peptides from Bcin06g02870, based on LiP-Quant score in the presence of increasing concentrations of BAYE-004. Bcin06g02870 is a serine/threonine kinase predicted to be casein kinase I. The extrapolated average EC50 for this protein is 6 nM. **d** Thermal stability of His-tagged Bcin06g02870 upon treatment with increasing concentrations of BAYE-004. Western blots and the corresponding quantification of the soluble fraction of Bcin06g02870 at 56 °C (n = 2). **e** Homology model of the structure of Bcin16g04330 kinase (from template PDBID: 3gzd, see Methods section). Red dot shows the center of mass of the LiP-Quant peptides (pink) for this protein. Magenta dots represent a 4 Å radius around the center of mass. It is positioned within the volume of the catalytic site (cyan), which is a common binding site of other kinase inhibitors. The volume of the catalytic site was calculated based on a model of a protein bound to a kinase inhibitor (PDBID: 3gzd). **f** LANCE Ultra kinase assay of CK-1 inhibition upon incubation with increasing concentrations of BAYE-004. Source data are provided as a Source Data file.

drug treatment represents a more biologically relevant experiment, while ensuring proteins maintain their native compartment during compound incubation. However, one must bear in mind that the ensuing cell lysis required for limited proteolysis could introduce artifacts, most likely driven by enabling novel protein–protein interactions or residual compound access to previously compartmentalized proteins as equilibration is re-established in the lysate. Despite this, using live-treated cells we robustly identify the known target of rapamycin, confirming that further steps towards an "in situ" application of LiP-based technologies are possible. As such approaches deviate from the standard LiP-Quant pipeline, they are in their infancy and will require further development to reduce false positives/negatives but they offer a glimpse of potential areas for further development.

Collectively, this work demonstrates that LiP-Quant effectively identifies protein–drug targets, characterizes their binding properties across species and drug target classes, including kinases, phosphatases, regulatory proteins and membrane proteins, and pinpoints drug binding regions of allosteric or catalytic nature. We have shown that the LiP approach is very robust in human cell lysates and can also be further developed in live cells, as well as to target plasma membrane proteins. Other chemoproteomics approaches such as TPP are routinely applied in vivo and in tissues[8] but they have lower structural resolution, as they measure variations of thermal stability of whole proteins instead of proteolytic susceptibility. We envision that TPP and LiP-Quant will be used in combination in the future to achieve a more comprehensive coverage of the druggable proteome. These capabilities make LiP-Quant a powerful target deconvolution strategy with the potential to become an essential part of the chemoproteomics toolbox and drug-discovery pipeline.

## Methods

**Experimental model and subject details.** *Saccharomyces cerevisiae* cells (Euroscarf, Supplementary Data 16) were grown at 30 °C in YPD media to early log phase from a single colony picked from a fresh YPD plate. Cells were harvested by centrifugation and carefully washed three times with ice-cold lysis buffer (100 mM HEPES pH 7.5, 150 mM KCl, 1 mM MgCl$_2$). Cell pellets were resuspended in lysis buffer, and cell suspensions were extruded from a gauge needle to produce drops that were immediately flash frozen in liquid nitrogen.

*Botrytis cinerea* (clone BO47), both wild-type and CK1 His-tagged, cells (Bayer Crop Science, Supplementary Data 16) were cultured in potato dextrose agar (39 g/L, Oxoid #CM0139) at 21 °C for 10 days. After 10 days growth the cells were suspended in 10 ml of GYPm liquid media (14.6 g/l D(+)-glucose monohydrate (VWR #24370.320), yeast extract (Merck #1.03753.0500), mycological peptone (Oxoid #LP0040)) and filtered (100 μm, Corning cell strainer) to harvest spores (final solution of $5 \times 10^6$ spores/ml). This liquid culture was incubated for 24 h at 110 rpm at 21 °C. Cell mycelium was pelleted by centrifugation (5 min, 16,000 g, 21 °C), media was removed and the pellet snap frozen in liquid nitrogen.

**Whole-proteome preparation for MS analysis.** *Saccharomyces cerevisiae*: Liquid-nitrogen frozen beads of *S.cerevisiae* cell suspensions in lysis buffer (100 mM HEPES pH 7.5, 150 mM KCl, 1 mM MgCl$_2$) were mechanically ground in cryogenic conditions with a Freezer Mill (SPEX SamplePrep 6875). Cell debris was removed by centrifugation (10 min, 20,000 g, 4 °C). The sample preparation procedure was performed at 4 °C.

HeLa and *Botrytis cinerea* cells (lysate): All biological samples were kept on ice through sample preparation. HeLa cell pellets (Ipracell, CC-01-10-10) ($5 \times 10^7$ cells) and *Botrytis cinerea* mycelium ($3 \times 10^7$ cells) were resuspended in 800 μl LiP buffer (100 mM HEPES pH 7.5, 150 mM KCl, 1 mM MgCl$_2$) and lysed by passing completely through a BD Precision glide syringe needle (27 G) ten times, followed by 20 min incubation on ice. Lysate was cleared by centrifugation (16,000 g at 4 °C) for 4 min. Supernatant was retained in a new Eppendorf tube and the pellet was resuspended in 400 μl of LiP buffer for repeated lysis under the aforementioned conditions, including incubation and centrifugation. After centrifugation, supernatants were combined and protein amount was determined using a Pierce BCA Protein Assay Kit (cat #23225) according to manufacturer's instructions.

HeLa cells (in vivo culture): HeLa cells (Sigma-Aldrich, 93021013-1VL) were cultured in low-glucose Dulbecco's Modified Eagle's Medium (DMEM) (Sigma-Aldrich, #D6046) supplemented with 10% fetal bovine serum (FBS) and 1% penicillin/streptomycin. Cells were passaged prior to confluency by detachment with 0.25% trypsin and subculture at a ratio of 1:8.

**Wheat germ agglutinin beads membrane protein enrichment.** HeLa cells were lysed as above (lysate) however 0.3% *n*-dodecyl-β-D-maltoside (DDM) was added prior to lysis. Four mg of lysate was rotated with 320 μl of wheat germ agglutinin agarose beads (Reactolab SA, AL-1023-2), pre-washed twice with LiP buffer, for 4 h at 4 °C. Beads were then centrifuged at 300g for 30 s to pellet and the supernatant was aspirated. Beads were then washed six times (300g, 30 s between washes) and finally eluted in 400 μl of 0.5 M N-acetyl glucosamine (in LiP buffer without detergent) at 4 °C for 30 min. Beads were pelleted at 300g for 30 s and eluate was collected and protein was quantified by BCA test. LiP-Quant assay was performed with membrane enriched eluate in the same manner as described for standard HeLa lysate.

**Lysate and cell treatment for LiP-Quant in native conditions.** *Saccharomyces cerevisiae*: Cell lysates from at least three independent biological replicates were aliquoted in equivalent volumes containing 100 μg of proteome sample and incubated for 10 min at 25 °C with the drug of interest. Proteinase K from *Tritirachium album* (Sigma Aldrich) was added simultaneously to all the proteome-metabolite samples with the aid of a multichannel pipette, at a proteinase K: substrate mass ratio of 1:100, and incubated at 25 °C for 4 min. Digestion reactions were stopped by heating samples for 5 min at 98 °C in a thermocycler followed by addition of sodium deoxycholate (Sigma Aldrich) to a final concentration of 5%. Samples were then heated again at 98 °C for 3 min in a thermocycler. These samples were then subjected to complete digestion in denaturing conditions as described below.

HeLa and *Botrytis cinerea* cells: 100 μg of protein lysate was aliquoted from a lysate pool for each of four independent replicates ($n = 4$ for all experiments) and incubated at room temperature (RT) with the compound of interest for 10 min. An 8-concentration dose–response was used for each experiment (seven compound dilutions plus a vehicle control) plus a single concentration rapamycin treatment as a positive control. For the rapamycin dose–response an additional concentration was used in place of the positive control. Proteinase K (1:100 ratio of enzyme to protein) was added and samples were incubated for a further 4 min. Samples were transferred to a heat block at 98 °C for 1 min, at which time proteinase K activity was quenched with an equal volume of 10% deoxycholate (to a final concentration of 5%) and incubated for a further 15 min at 98 °C.

HeLa cells: Near confluent 6-well plates (9.6 cm$^2$ per well) were washed twice with DMEM minus FBS, followed by incubation with rapamycin (2 μM) or DMSO (0.2%) ($n = 3$ for each of biologically independent wells of cells) in DMEM minus FBS at 37 °C for 15 min. At the end of compound incubation cells were washed twice with ice-cold LiP buffer and then scraped into an Eppendorf tube in 100 μl of LiP buffer, which was immediately snap frozen. Cells were thawed at 4 °C and snap frozen again for a total of three freeze-thaw cycles. After the final thaw, proteinase K (1 μg per well) was added and samples were incubated at room temperature for 4 min. Samples were transferred to a heat block at 98 °C for 1 min, at which time proteinase K activity was quenched with an equal volume of 10% deoxycholate (to a final concentration of 5%) and incubated for a further 15 min at 98 °C.

**Proteome preparation in denaturing conditions.** Samples were removed from heat and reduced for 1 h at 37 °C with 5 mM tris(2-carboxyethyl)phosphine hydrochloride followed by a 30 min incubation at RT in the dark with 20 mM iodoacetamide. Subsequently, samples were diluted in two volumes of 0.1 M ammonium bicarbonate (final pH of 8) and digested with lysyl endopeptidase (1:100 enzyme: substrate ratio). Samples were further digested for 16 h at 37 °C with trypsin (1:100 enzyme: substrate ratio). Deoxycholate was precipitated by addition of formic acid to a final concentration of 1.5% and centrifuged at 16,000g for 10 min. After transferring the supernatant to a new Eppendorf tube an equal volume of formic acid was added again and the centrifugation repeated. Digests were desalted using C18 MacroSpin columns (The Nest Group), or Sep-Pak C18 cartridges or into 96-well elution plates (Waters), following the manufacturer's instructions and after drying resuspended in 1% acetonitrile (ACN) and 0.1% formic acid. The iRT kit (Biognosys AG, Schlieren, Switzerland) was added to all samples according to the manufacturer's instructions.

**High pH reversed phase fractionation.** Equal amounts of peptides were taken and pooled from the final LiP reaction digests for each treatment (e.g. 7 μg from each replicate for each condition), resulting in approximately 200 μg of total digest. This digest pool was fractionated into 10-12 fractions using high pH reversed phase chromatography with a Dionex Ultimate 3000 HPLC (Thermo Fisher, Waltham, United States) and an ACQUITY UPLC CSH C18 column (1.7 μm × 150 mm) from Waters (Milford, United States). In brief, a 25% ammonium hydroxide solution was used to adjust the pH of the digest pool to 10. The lysate was run on a 30 min non-linear gradient, increasing from 1 to 40% ACN, at a flow rate of 0.3 ml per min and a micro-fraction size of 30 s. After drying the individual fractions were resuspended in 1% ACN and 0.1% formic acid and Biognosys' iRT kit was added.

**Mass spectrometric acquisition.** For all samples generated from HeLa or *Botrytis cinerea* cells, for DIA (Data Independent Acquisition) runs, 2 μg of LiP reaction digest from each sample was analyzed using an in-house analytical column (75 μm × 50 cm). Samples were block randomized before acquisition. PicoFrit

PicoTip Emitters (SELF/P Tip 10 μm) were packed with ReproSil-Pur C18-AQ 1.9 μm phase (Dr. Maisch, Ammerbuch-Entringen Germany) and connected to an Easy-nLC 1200. All experiments were run on a Q-Exactive HF mass spectrometer (Thermo Scientific) with the exception of the calyculin A dataset, which was acquired on a Q-Exactive HF-X. Peptides were separated by a 2 h segmented gradient at a flow rate of 250 nl/min with increasing solvent B (0.1% formic acid, 85% ACN) mixed into solvent A (0.1% formic acid, 1% ACN). Solvent B concentration was increased from 1% after 3 min according to the following gradient: 4% over 3 min, 5% for 3 min, 7% for 4 min, 9% for 5 min, 11% for 8 min, 16% for 19 min, 26% for 41 min, 29% for 9 min, 31% for 6 min, 33% for 5 min, 35% for 4 min, 38% for 4 min, 40% for 3 min, 44% for 3 min, 55% for 3 min and 90% in 10 s. This final concentration was held for 10 min followed by a rapid decrease to 1% over 10 s, which was then held for 5 min to finish the gradient. A full scan was acquired between 350 and 1650 $m/z$ at a resolution of 120,000 (ACG target of 3e6 or 7 ms maximal injection time). A total of 37 DIA segments on HF were acquired at a resolution of 30,000 (ACG target of 3e6 or 47 ms maximal injection time) and 42 on the HF-X (ACG target of 3e6 or 55 ms maximal injection time). The normalized collision energy was stepped at 25.5, 27, 30. First mass was fixed at 200 $m/z$.

For DDA (Data Dependent Acquisition) runs from the same samples, peptides were separated by the same 2 h segmented gradient as utilized above for DIA runs with the exception that the final 1% solvent B flow was held for 4 min and 40 s (rather than 5 min). All experiments were run on a Q-Exactive HF mass spectrometer (Thermo Scientific) with the exception of the rapamycin (Q-Exactive HF-X) and FK506 datasets (Q-Exactive). A top 15 method was used across a scan range of 350 to 1650 $m/z$ with a full MS resolution of 60,000 (ACG target of 3e6 or 20 ms injection time). Dependent MS2 scans were performed with a resolution of 15,000 (ACG target of 2e6 or 25 ms injection time) with an isolation window of 1.6 $m/z$ and a fixed first mass of 120 $m/z$.

Peptide samples generated from *Saccharomyces cerevisiae* were analyzed on an Orbitrap Q Exactive Plus mass spectrometer (Thermo Fisher Scientific) equipped with a nano-electrospray ion source and a nano-flow LC system (Easy-nLC 1000, Thermo Fisher Scientific). MS data acquisition in DDA and DIA modes was essentially carried out as in Piazza et. al. 2018.

**Mass spectrometric data analysis**. DIA spectra were analyzed with Spectronaut X (Biognosys AG)[40] using the default settings. In brief, retention time prediction type was set to dynamic iRT (adapted variable iRT extraction width for varying iRT precision during the gradient) and correction factor for window 1. Mass calibration was set to local mass calibration. The false discovery rate (FDR) was estimated with the mProphet approach[41] and set to 1% at both the peptide precursor and protein level. Statistical comparisons were performed on the modified peptide level using fragment ions as quantitative input. The DDA spectra were analyzed with the SpectroMine (Biognosys AG) software using the default settings with the following alterations. Digestion enzyme specificity was set to Trypsin/P and semi-specific. Search criteria included carbamidomethylation of cysteine as a fixed modification, as well as oxidation of methionine and acetylation (protein N-terminus) as variable modifications. Up to 2 missed cleavages were allowed. The initial mass tolerance for the precursor was 4.5 ppm and for the fragment ions was 20 ppm. The DDA files were searched against the human UniProt fasta database (updated 2018-07-01) and the Biognosys' iRT peptides fasta database (uploaded to the public repository). The libraries were generated using the library generation functionality of SpectroMine with default settings.

**Machine learning-based training of the LiP-Quant classifier**. All HeLa datasets were first analyzed for differentially regulated peptides between the highest drug concentration and vehicle using Spectronaut's statistical testing (one sample two-sided *t*-test with Storey method correction) performed on the modified peptide sequence level using fragment ions as the smallest quantitative units. This candidate peptide list was filtered based upon *q*-value < 0.01 and an absolute log₂ fold-change > 0.58. Each peptide in this filtered list was then subjected to dose–response correlation testing (using the drc package (https://www.r-project.org)) on all peptides (modified sequence with fragments ions as quantitative units) at every drug concentration to establish a sigmoidal correlation coefficient.

As the ground truth (target proteins) was known for the drugs tested in HeLa lysates, each protein identified in each dataset was annotated as either a known target or non-target and from this a contaminant database, or LiP-protein frequency library (PFL), was built. To do so, the same statistically filtered list of differentially regulated peptides as above was used and proteins that were present but not specific for the drug being tested were quantified and assigned a PFL (contamination) score. For example, a protein that showed differential regulation in 9 of 11 ground truth experiments (several experiments were performed more than once) was assigned a contamination score of 9/11 or 81.8% (Supplementary Data 14), proteins that never appeared as contaminants in any experiment were not included in the PFL-library. This library enabled the quantitative down-weighting of proteins that were frequently present in LiP experiments but not specific for the drug being tested. We observed high correlation between proteins identified as likely contaminants in the PFL of our LiP-Quant experiments (Supplementary Data 14) and those previously identified as common contaminants in affinity purification mass spectrometry (such as chaperone and structural proteins)[42].

To establish the criteria that contribute to the identification of drug targets, we split our dose–response experimental data (filtered based on q-value and log₂ fold-change and PFL annotated as mentioned above) into two independent datasets to train our classifier; training set A included the drugs calyculin A, rapamycin and staurosporine and training set B included FK506, selumetinib and fostriecin (Supplementary Fig. 4A). For each training set the data was combined and we used linear discriminant analysis (LDA) to build classifiers based upon all potential unique peptide/protein features (e.g. dose–response correlation, PFL frequency, protein coverage, etc). For each training set, known drug targets were selected as a positive training set, resulting in 95 modified sequences for training set A and 33 for training set B. We also randomly sampled 400 background modified sequences as a negative training set from each training set. The features were calculated and stabilized to a defined range between 0 and 1. The LDA-based machine learning was performed five times for each training dataset with resampling of the negative training set each time. The identified criteria were consistent across all LDA analyses (Supplementary Fig. 1B) and the contribution weights for each of the features from the five LDA analyses was averaged. The relative contributions of each parameter to the LiP-Quant score was very stable across the training sets (Supplementary Fig. 4B). We termed the linear classifier the LiP-Quant Score in this study. The weights were adjusted such that the combined linear classifier could reach a maximum value of 6. These weightings were incorporated into the analysis pipeline (see below) and verified independently on the other positive control datasets (i.e. training set A was verified on the datasets comprising training set B and vice versa) (Supplementary Fig. 4A). LiP rankings using both training set analysis parameters were similar across all datasets (Supplementary Fig. 4C).

Using this approach, we established four classifiers that contribute to positive drug target identification (Supplementary Fig. 1B): (I) correlation of fit with a dose–response binding model, (II) the presence of the identified protein in the LiP-protein frequency library, (III) the number of peptides from an identified protein showing regulation that are in the top ten percent of all peptides ranked by q-value in the Spectronaut filtered statistical test (see above) and (IV) the statistical significance (q-value) of the relative peptide abundances between drug and vehicle-treated samples. As training set A contained a larger positive training set (i.e. there were more known drug target peptides identified) the weightings calculated for this training set were used for all subsequent analyses.

**Automated peptide/protein ranking of LiP-Quant experiments**. Using the criteria and weightings established from our training datasets we wrote in-house scripts in R to calculate in an unbiased manner the individual peptide sub-scores for each LiP-Quant experiment. As these experiments contained on average over 100,000 peptides, peptides were first filtered based upon differential abundance from the Spectronaut statistical testing table (one sample two-sided *t*-test with Storey method correction, q-value < 0.01 and an absolute log₂ fold-change > 0.46) using statistical comparisons against vehicle control for a range of drug concentrations (IC₅₀ through 1000-fold the IC₅₀, or the range closest to this). Each peptide in this narrowed down putative candidate list was then subjected to full LiP-Quant analysis using the four weighted criteria (Supplementary Fig. 1B) described above and a final LiP-Quant score for each peptide was calculated.

This final analysis pipeline enabled the selection and ranking of the most relevant peptides and proteins per experiment. The combined LiP-Quant score enables direct comparison of LiP peptides with each other and allows more robust discrimination of genuine targets from random hits. Ranking on the protein level was performed using the best LiP-score per protein, only. All half maximal effective/inhibitory concentrations (EC₅₀/IC₅₀) were calculated using the drc package (https://www.r-project.org). The necessary output files from Spectronaut are outlined in the docstring at the start of the R script.

**Criteria used for establishing a LiP threshold score**. Aggregating results from five positive control experiments (rapamycin, calyculin A, selumetinib, FK506 and fostreicin) conducted in HeLa lysate and analyzed with our LiP-Quant pipeline, we found that LiP scores show a bimodal distribution. Staurosporine was excluded from the threshold calculation as it shows a level of promiscuity (binding potentially hundreds of kinases) that is rare among drugs, making it difficult to ascertain if low scoring peptides are genuine targets or kinases that were not detected or kinases that are not bound by the drug. As this difficulty in interpreting non-target peptides could bias the threshold calculation the dataset was excluded. Peptides from known target proteins show a clear enrichment in the high-scoring peak of the distribution (LiP-Quant score > 1.5), whereas all other peptides are enriched in the low-scoring peak of the distribution with a median of approximately 0.8 (Fig. 1b). We defined a threshold score of 1.5 by taking the median LiP-Quant score from the aforementioned experiments, plus three standard deviations, to ensure minimal (<1%) non-target peptide presence (Supplementary Data 3). Although the approach ensures a strong enrichment for genuine targets, it should be noted that some peptides from these targets are expected below a LiP-Quant score of 1.5 as both LiP-Quant and non-LiP-Quant peptides can be expected from genuine target proteins.

**Guidelines for interpreting LiP-Quant results**. The purpose of the LiP-Quant score is to provide a candidate list of protein (and peptide) targets ranked by their likelihood of being a genuine drug interactor. The LiP-Quant scoring system covers

the range of 0 to 6, with a peptide scoring 6 having the maximum probability of being a true target. In this way, these scores also assign ranks to proteins, by way of their peptides, enabling an unbiased prioritization of potential targets. This absolute scale is useful to make direct comparisons between experiments. For instance, the strongest peptide and protein candidate targets in LiP-Quant experiments are typically those with a score > 2.5 and rank in the top-scoring peptides of the whole proteome. Every LiP-Quant peptide has an EC50 value assigned, which corresponds to the inferred dose of drug necessary to observe half-maximum of the relative peptide intensity variation between drug and vehicle. This could also be, in principle, used as a discriminating factor under the assumption that drug targets with low EC50 values should be indicative of a strong binding interaction between protein and compound, and a compound that binds with high affinity (e.g. nM or lower) is more likely to be an effective drug in vivo. However, we normally do not exclude candidates with EC50 close to the uM range a priori, as compounds that weakly bind the target and have a phenotypic effect can be further refined during pre-clinical drug development.

**Structural models**. The amino acid conservation in the structural model of calyculin A bound to the PP1-gamma catalytic subunit has been calculated using the ConSurf algorithm (Landau 2005). Structures of *Botrytis cinerea* protein–drug targets were modeled using homolog kinases with high sequence similarity for which experimental structural data was available using the Swiss-model. The *Botrytis cinerea* kinase models were then structurally aligned: Bc Bcin06g02870 with human kinase CSNK1A1 bound to the kinase inhibitor A86 (PDBID 6gzd) and *Botyrris cinerea* Bcin16g04330 with human kinase Abl in complex with imatinib and GNF-2 (PDBID 3k5v). Given the high structural and sequence similarity between *Botrytis* and human kinases, we used the experimental data relative to the holocomplexes between kinase inhibitors and protein kinases to assign the position of the ATP-binding (catalytic) site and allosteric sites of the *Botrytis* kinases.

**Definition of PPV**. We defined the PPV as the ratio between the number of true positive peptides and the sum of false positives (FP) and true positives (TP) identified by LiP-Quant (TP/ (TP + FP)). Unless specifically stated, these parameters refer to the top 50 LiP-Quant score ranking peptides.

**Benchmarking of the LiP-Quant classifier**. The LiP-Quant staurosporine experiment was selected to provide an estimation of the method FDR because of the known binding promiscuity of the compound. Published datasets of staurosporine proteome-wide binding profiles obtained with TPP[6] and kinobeads[23] were used to compare the predictive power of the three chemoproteomic methods. We used LiP-Quant scores for LiP-Quant, -log$_{10}$(adjusted *p*-values) of the $R^2$ correlation indices of the melting curves for TPP reporting both replicates analyzed in the original publication[6], and the number of spectral counts (PSM) for kinobeads as ranking criteria. For the comparison of LiP-Quant, TPP and kinobeads in the venn diagram of Fig. 2b, we considered as TPP and kinobeads hits, those proteins defined as hits by the authors in the original publications. For kinobeads, these were the isolated proteins in the staurosporine-beads pull down. For TPP, Savitski et al.[6] defined protein hits as those fulfilling the following criteria: (I) the minimal slope is below −0.06 in both biological replicates; (II) the minimal differential melting temperature in experiment 1 and 2 is higher than the same difference measured in the corresponding experiments with vehicle; (III) the differential melting temperature in experiment 1 and 2 have the same sign; (IV) the adjusted p-values of the $R^2$ correlation indexes are below 0.05 in experiment 1 and below 0.1 in experiment 2 or the adjusted p-values of the $R^2$ correlation indexes are below 0.1 in experiment 1 and below 0.05 in experiment 2.

**Approximating drug binding sites from LiP-Quant data**. We validated our predictive strategy to estimate the position of drug binding sites with the LiP-Quant experiments for which true targets are known (Fig. 3) and consider only the protein hits that have multiple high-scoring LiP-Quant peptides. We chose the top 3 LiP-Quant peptides of a protein–drug target candidate among the 15 highest-ranking peptides by LiP-Quant score of the whole proteome (Supplementary Data 15). Only one protein target candidate typically fulfilled these criteria in all analyzed experiments. LiP-Quant peptides measured with rapamycin, FK506, selumetinib, staurosporine, fostriecin and calyculin A in HeLa proteome extracts were analyzed. We analyzed the staurosporine dataset assayed using a 4 h long LC gradient. We calculated the position of drug binding sites using the center of mass of all atoms assigned to the aforementioned top 3 LiP-Quant peptides of the main candidate target. Structural models and geometric calculations were performed using PyMol 2.1.1 (Schrodinger).

**Cellular thermal shift assay (CETSA)**. *Botrytis cinerea* BO47 (CK1 His-Tagged) cell suspension was adjusted to 1 × 10$^6$ sp/ml GYPm and incubated for 24 h at 21 °C (110 rpm). 12.5 × 10$^6$ cells were treated with BAYE-004 (at various concentrations from 0.0001 to 67.5 μM) or control (1% DMSO) for the final 20 min of the 24 h growth period. Cells were harvested by filtration (100 μm) and rinsed with 15 ml of ice-cold HEPES buffer (0.1 M HEPES, 50 mM NaCl, pH 7.5). Harvested mycelium was resuspended in 3.5 ml HEPES buffer and kept on ice. 500 μl of each

concentration was transferred to a 2 ml Eppendorf tube and heated to 56 °C on a thermoshaker for 3 min, an additional aliquot from each concentration was left unheated. After heating, cells were kept on ice for 3 min, snap frozen in liquid nitrogen, lyophilized overnight and then stored at −80 °C until protein extraction.

Lyophilized mycelium was lysed using a Retsch mixer mill (MM 400) with 3 mm tungsten carbide beads (30 Hz for 3 s, two cycles), then 500 μl of cold protein extraction buffer (50 mM HEPES, 50 mM NaCl, 0.4% NP-40) was added. Lysate was incubated for 10 min at 25 °C, centrifuged (10 min, 14,000g) and the supernatant was retained. The lysate was further centrifuged (20 min, 73,400g) to eliminate insoluble proteins. The supernatant was collected, and protein concentration was determined using the Qubit protein assay kit (#Q33211) and stored at −20 °C.

Target engagement was assessed by western blot. In brief, 17 μg of protein per treatment was loaded onto a TGX (4-20%) stain free gel (Bio-Rad, #4568094) and run at 250 V for 25 min. Proteins were transferred to a nitrocellulose membrane using the Trans-Blot Turbo system according to the manufacturer's instructions (Bio-Rad, # 1704271). The membrane was probed using a monoclonal anti-polyhistidine-peroxidase antibody (1:2000, clone HIS-1, Sigma, A7058). The membrane (target protein) and gel (loading control) were imaged using a ChemiDocXRS camera and quantified using ImageJ[43]. The uncropped blot image is included in the source data file (see Data Availability).

**Cell viability (IC50) assay**. *Botrytis cinerea* BO5.10 (2 × 10$^3$ cells/ml) mycelium in GYPm liquid media (200 μl) was cultured at 21 °C without shaking in a micro-titer plate. Optical density was measured at 620 nm (Tecan M1000 plate reader) at the beginning of the culture period (day 0) and immediately inoculated with 2 μl of BAYE-004 to obtain final concentrations (μM) of 1.2234, 0.40745, 0.13582, 0.04527, 0.01509, 0.00168, 0.00056, 0.00019, and 0, respectively. The culture was grown for three days at 21 °C after inoculation at which point the optical density was measured again. Inhibition of cell growth was calculated at each concentration.

**CK1 kinase assay**. The LANCE Ultra time resolved fluorescence resonance energy transfer (Tr-FRET) kinase assay (Perkin Elmer, #TRF0300-C) protocol was used according to manufacturer's instructions with adaptations made for the following conditions. *Botrytis cinerea* MBP-CK1 recombinant enzyme (89.7 kDa) was used at 12.5 nM with the following reagents: ULight-DNA topoisomerase 2-α(Thr1342) peptide (Phosphorylation motif: DEKTDDE, PerkinElmer, #TRF0130-M), Europium-labeled DNA topoisomerase 2-α(Thr1342) antibody (mouse monoclonal, PerkinElmer, #TRF0218-M) in a solution containing DMSO (1%). The reaction was performed in duplicates in the dark at 30 °C for 90 min. Plates were read in a Victor2 Perkin plate reader (excitation 340 nm/emission 665 nm).

**Materials**. Details of all materials used in these studies are provided in Supplementary Data 16. All chemicals, enzymes, peptides and compounds were purchased from Sigma-Aldrich unless specified otherwise. BAYE-004 was produced by Bayer Crop Science (Supplementary Note 1 and Supplementary Figs. 10, 11). MBP-CK1 recombinant enzyme was cloned in-house by Bayer Crop Science. Frozen HeLa cell pellets were purchased from Ipracell (Belgium) and live HeLa cells for culture were purchased from Sigma-Aldrich. Strains of *S.cerevisiae* were obtained from the European Saccharomyces Cerevisiae Archive for Functional Analysis (Euroscarf) or subcloned from them (Supplementary Data 16). B. Cinerea was provided by Bayer Crop Science (Supplementary Data 16), over-expression construct for CK1-His was cloned in-house by Bayer Crop Science. All software versions used and where they were obtained is outlined in Supplementary Data 16.

**Reporting summary**. Further information on research design is available in the Nature Research Reporting Summary linked to this article.

## Data availability

All mass spectrometry proteomics data have been deposited at ProteomeXchange Consortium via the PRIDE partner repository with the dataset identifiers PXD018204 and PXD015446. The source data underlying Figs. 1a, 2, 4b, d, f, and Supplementary Figs. 2b, c, 3, 3b, e, 6a, b and 8a are provided as a Source Data file. Source data underlying Figs. 1d, 2c, d, 4c, and Supplementary Figs. 4a, b, e, 5c, 7a, b and 9a are included in the PRIDE repository with the dataset identifier PXD019902. All other relevant data are available from the corresponding authors on request.

UniProt fasta databases for organisms were accessed on January 1st, 2018 via the UniProt databases download page (https://www.uniprot.org/downloads). Protein structures for E2AK2 kinase (PDBID: 2a19[25]), MAPK2K1 (PDBID: 4u7z[44]), K66A3 kinase (PDBID: 4nus[45]), FKBP1A (FK506) (PDBID: 1fkj[46]), FKBP1A (rapamycin) (PDBID: 2dg3[47]) and PP2A (PDBID: 1it6[48]) were downloaded in 2019 from the Protein Data Bank website (https://www.rcsb.org/pdb).

## Code availability

The custom R script used to compute LiP scores and ranks is available via GitHub at https://github.com/RolandBruderer/LiP-Quant.

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

## Acknowledgements

The authors thank the Haering laboratory (EMBL Heidelberg) for providing yeast strains. P.P. is funded by a Personalized Health and Related Technologies (PHRT) grant (PHRT-506), a European Research Council Consolidator (ERC-CoG) grant (ERC-CoG 866004) and a Sinergia grant from the Swiss National Science Foundation (SNSF grant CRSII5_177195). P.P. receives funding from the European Union's Horizon 2020 research and innovation programme (ERC grant 866004 and also grant 823839), a Personalized Health and Related Technologies (PHRT) grant (PHRT-506) and a Sinergia grant from the Swiss National Science Foundation (CRSII5_177195).

## Author contributions

L.R., P.P., N.B., and I.P. conceived and designed the study. L.R., N.B., I.P., and T.K. designed experiments. N.B. and I.P. performed all limited proteolysis experiments. C.B., L.C., A.S., and I.S. performed other experiments. N.B., I.P., R.B., and T.K., analyzed data. I.P. and N.B. wrote the manuscript with contributions from R.B., T.K., O.R., N.dS., P.P., and L.R.

## Competing interests

I.P., N.B., R.B., O.R., and L.R. are employees of Biognosys AG. O.R. is a co-founder of Biognosys AG. P.P. is a scientific advisor for the company Biognosys AG (Zurich, Switzerland) and inventor of a patent licensed by Biognosys AG. T.K., A.S., C.B., and L.C. are employees of Bayer SAS (France). I.S. is an employee of BASF SE. The remaining authors declare no competing interests.
