## [Peer Review File · Nature Communications]

Reviewers' Comments:

Reviewer #2:

Remarks to the Author:

In this work, Piazza and Beaton et al. improve their previous limited proteolysis approach to identify targets of small molecules in cell lysates. Compared to previous studies, the authors used a dose-response approach that allows the use of a more stringent statistical method. I had previously reviewed this manuscript for another journal and most of my comments have been addressed. This has significantly improved the manuscript, but there are still some points that I think the authors need to address.

1. The authors should address the limitations of their intact cell experiment, since limited proteolysis needs to be performed after lysis (even if the treatment is performed in living cells). The authors should include in the discussion that the lysis procedure can lead to reequilibration of compound binding and protein-protein interactions. This can lead to the compound no longer preventing proteolysis of some proteins, or the compound being able to access proteins that were not available before (e.g., due to subcellular compartmentalization) – leading to possible false negatives and positives.

2. While I appreciate that the authors now cite more recent and sensitive chemical biology approaches, the comparison of their method to these is still not entirely fair.

- a. First, only one replicate from the Savitski 2014 dataset was used. It is clear from the publication from which this dataset was retrieved that in this mode of TPP (TPP-TR), at least two replicates of each condition should be used.

- b. Second, the authors did not properly annotate the TPP dataset, probably due to problems matching IPI access numbers to Uniprot IDs – a quick look reveals that the third hit and some hits further down (LIMK2, STK4 and STK11) are not annotated as kinases despite being present in kinhub.org (and the authors Table S6). This leads to these proteins being annotated as false positives.

- c. This led me to repeat the benchmark, by using the same datasets (Savitski 2014, LiP-Quant and Deep-LiP_Quant), but with a more sensitive way of calling hits that was used to analyze this staurosporine TPP dataset and published last year (Childs et al. Mol. Cell. Prot. 2019) and carefully annotating all the datasets. I attach a markdown file of this analysis that I am happy that the authors use directly in their manuscript. This highlights that, as pointed out in my first comments, LiP (and also Deep-LiP) has lower coverage than other chemical biology approaches and is therefore less sensitive. There is no need for the authors to try to hide that LiP-Quant is less sensitive than other approaches, since it has other strengths, such as the ability to pinpoint the binding site of the compound.

REVIEWER COMMENTS

We thank both Dr. Loessl and the reviewer for their constructive feedback. We believe that the revisions (highlighted in yellow) that we have made based upon their suggestions and insights have further improved the quality of our manuscript. We have addressed the concerns of the reviewers as follows:

Reviewer #2 (Remarks to the Author):

In this work, Piazza and Beaton et al. improve their previous limited proteolysis approach to identify targets of small molecules in cell lysates. Compared to previous studies, the authors used a dose-response approach that allows the use of a more stringent statistical method. I had previously reviewed this manuscript for another journal and most of my comments have been addressed. This has significantly improved the manuscript, but there are still some points that I think the authors need to address.

1. The authors should address the limitations of their intact cell experiment, since limited proteolysis needs to be performed after lysis (even if the treatment is performed in living cells). The authors should include in the discussion that the lysis procedure can lead to reequilibration of compound binding and protein-protein interactions. This can lead to the compound no longer preventing proteolysis of some proteins, or the compound being able to access proteins that were not available before (e.g., due to subcellular compartmentalization) – leading to possible false negatives and positives.

We agree with the reviewer that a degree of protein rearrangement, as well as decompartmentalization, is inevitable in our intact drug-treated cells and that this could introduce false positive and/or false negative target identifications. We have expanded our discussion to reflect these possibilities and more accurately reflect the scope of this protocol. Changes to the manuscript are highlighted in the discussion.

2. While I appreciate that the authors now cite more recent and sensitive chemical biology approaches, the comparison of their method to these is still not entirely fair.

a. First, only one replicate from the Savitski 2014 dataset was used. It is clear from the publication from which this dataset was retrieved that in this mode of TPP (TPP-TR), at least two replicates of each condition should be used.

We now present the results of both biological replicates reported in the 2014 Savitski publication in **Figure 2A**. Since Savitski *et al.* computed separate t-tests for differential melting temperatures for each replicate, we show independent curves for each replicate. We acknowledge that the sensitivity of TPP for true protein targets appears to be higher than that of LiP-Quant specifically between the top 25 – top 50 ranking protein candidates for the second staurosporine replicate (**Figure 2A**, TPP-replicate 2). To remove any possible residual bias, we therefore removed the following statement from the main text: “LiP-Quant discriminates true positive targets better than TPP as LiP-Quant prioritize more protein kinases in the amongst the top ranking protein candidates”.

b. Second, the authors did not properly annotate the TPP dataset, probably due to problems matching IPI access numbers to Uniprot IDs – a quick look reveals that the third hit and some hits further down (LIMK2, STK4 and STK11) are not annotated as kinases despite being present in kinhub.org (and the authors Table S6). This leads to these proteins being annotated as false positives.

The discrepancies arise from the fact that the International Protein Index (IPI) format has been obsolete since 2011 and is no longer officially curated since then. In the previous version of this manuscript, we converted IPI identifiers using the last officially released version available here (<https://www.ebi.ac.uk/IPI>). In order to remove the missing values we have now used Biomart and the AnnotationDbi package from Bioconductor to make the mapping more complete. We have manually double checked the quality of the re-mapped matches, and verified that the cases pointed out by the reviewer are not anymore annotated as false negatives. We have also reanalyzed all data shown in **Figure 2A, 2B** and **2D** after correcting the annotation and reported the updated results, which do not radically change compared with the previous version. We emphasize that the IPI protein identifier format is no longer an officially supported standard, so small ambiguities are still possible, depending on the conversion method used.

c. This led me to repeat the benchmark, by using the same datasets (Savitski 2014, LiP-Quant and Deep-LiP_Quant), but with a more sensitive way of calling hits that was used to analyze this staurosporine TPP dataset and published last year (Childs et al. Mol. Cell. Prot. 2019) and carefully annotating all the datasets. I attach a markdown file of this analysis that I am happy that the authors use directly in their manuscript. This highlights that, as pointed out in my first comments, LiP (and also Deep-LiP) has lower coverage than other chemical biology approaches and is therefore less sensitive. There is no need for the authors to try to hide that LiP-Quant is less sensitive than other approaches, since it has other strengths, such as the ability to pinpoint the binding site of the compound.

In the previous version of this manuscript, we already acknowledged that: *“TPP identifies more protein targets for staurosporine than LiP-Quant (21 vs 45)”*. We have now edited the text with the following statement to make this point more explicit: *“Overall, the number of true positive targets found by the LiP-Quant and TPP methods are comparable, although TPP is more sensitive than LiP-Quant as it identifies more kinase targets in total.”*

We do wish to point out that a potential factor that enables higher kinase identification via TPP is cell line utilization. An in-house analysis based upon kinase expression levels from Pax DB shows a higher level of kinase expression in K652 cells (TPP) versus HeLa cells (LiP-Quant) (figure right). Based upon our data with Deep LiP-Quant it would be logical to conclude that our results would improve given further experiments in the same cell line. Unfortunately, such experiments are not currently possible in a timely fashion but can help to explain some degree of the sensitivity discrepancy observed between LiP-Quant and TPP.

We thank the reviewer to pointing out the Childs et al. publication, which was released during the preparation of this manuscript, and reports a new statistical analysis strategy based on non-parametric analysis of TPP dose response curves. We now perform a staurosporine comparative analysis among LiP-Quant, Deep-LiP, non-parametric TPP analysis (data from Childs et. al), and ‘standard’ Tm based TPP analyses (from both replicates reported in Savitski et. al. 2014) in **Figure 2D**. We also observe that the combination of TPP with non-parametric statistical analysis is the best predictor for staurosporine targets among the five approaches tested (Figure 2D, area under the curve = AUC = 0.85). We acknowledge this in the main text by adding the sentence: *“As recently reported, analyzing TPP data with a non-parametric approach increases specificity and sensitivity of*

TPP assays. We tested TPP in combination with non-parametric statistical analysis and observed that this has the highest predictive power for staurosporine targets...” and by removing any reference of LiP having better predictive power than TPP.